# An IoT-Based System for Monitoring the Health of Guyed Towers in Overhead Power Lines

**DOI:** 10.3390/s21186173

**Published:** 2021-09-15

**Authors:** Claudio Ferreira Dias, Juliane Regina de Oliveira, Lucas D. de Mendonça, Larissa M. de Almeida, Eduardo R. de Lima, Lucas Wanner

**Affiliations:** 1Instituto de Pesquisas Eldorado, Campinas 13083-898, Brazil; lucas.mendonca@eldorado.org.br (L.D.d.M.); eduardo.lima@eldorado.org.br (E.R.d.L.); 2Institute of Computing, University of Campinas, Campinas 13083-859, Brazil; juliane.oliveira@ic.unicamp.br (J.R.d.O.); lucas@ic.unicamp.br (L.W.); 3Taesa S.A., Rio de Janiero 20010-010, Brazil; larissa.almeida@taesa.com.br

**Keywords:** guyed transmission lines, tower collapse, Internet of Things, remote monitoring

## Abstract

The collapse of overhead power line guyed towers is one of the leading causes of power grid failures, subjecting electricity companies to pay considerable, high-value fines. In this way, the current work proposes a novel and complete framework for the remote monitoring of mechanical stresses in guyed towers. The framework method comprises a mesh network for data forwarding and neural networks to improve the performance of Low-Power and Lossy Networks. The method also considers the use of multiple sensors in the sensor fusion technique. As a result, the risk of collapse of guyed cable towers reduces, due to the remote monitoring and preventive actions promoted by the framework. Furthermore, the proposed method uses multiple input variable fusions, such as accelerometers and tension sensors, to estimate the tower’s displacement. These estimations help address the structural health of the tower against failures (i.e., loosening of the stay cables, displacement, and vibrations) that can cause catastrophic events, such as tower collapse or even cable rupture.

## 1. Introduction

The collapse of overhead power line guyed towers is one of the leading causes of power grid failures, subjecting electricity companies to pay considerable high-value fines [1,2]. In addition, city blackouts resulting from the collapse of cable-stayed structures cause a financial loss in the order of millions of dollars for companies and industries. Current traditional solutions, such as ground inspection by foot, seem ineffective since in-site periodic inspections are sometimes not enough to detect hidden defects in such structures [3]. Furthermore, the interval between visits and expensive personnel allocation makes monitoring/predicting defects difficult. They are not real-time, making it challenging to anticipate actions and prevent failures before the transmission tower’s tilt occurs.

The current work proposes a novel and complete framework for remotely monitoring mechanical stresses and the deformation of guyed towers. Specifically, the investigation focuses on the typical guyed-V steel lattice tower commonly seen in the Brazilian power grid, as depicted in Figure 1. The mechanical structure is also known as delta pylon, which supports high-voltage electric power transmission lines. The pylon has a V-shaped top for the cross beam admission and is usually established for one and, occasionally, for two electric circuits. Thus, these towers are for long-distance high voltage transmission in a 3-phase Delta configuration, requiring only three cables.

The overall framework idea is to envisage two ways of monitoring environmental threats. First, the remotely recorded data analysis allows finding evidence of risks in the guyed tower structure, due to long-term nature action. For example, it is possible to spot guyed tower structural failure from the data history, due to the correlation between harsh environment events and the guyed tower health. The instances of harsh environment events are intense temperature variation, strong wind, and heavy rain. Second, the constant online sampling of data in the cable traction allows real-time monitoring of the guyed tower health, which helps to prompt personnel to act immediately.

Consequently, a more effective corrective action helps the maintenance teams improve anchorages, increasing system reliability and the quality of service delivery. Furthermore, the reduction of in-field inspections and the adoption of ubiquitous monitoring ensure the wellness of the segment and improved quality of the energy service supply to the customer. Therefore, we expect the risk of collapse of guyed cable towers to reduce when the proposed framework is applied. In addition, we hope to facilitate inspection since remote monitoring allows to switch from periodic maintenance visits to on-demand maintenance visits, thus benefiting the electricity transmission segment.

The overall research work contributions are as follows: First, it offers a non-exhaustive overview on the state of the art of guyed tower remote monitoring. Second, the proposed framework allows addressing risks of collapse in guyed tower structures. Third, the method can consider real-time monitoring or historical data processing, allowing instant and preventive actions against disasters. Fourth, the proposal is an all-purpose solution by including a flexible and energy-saving mesh network and very different sensors. Fifth, network nodes are planned to have enough processing power to implement custom algorithms to improve data forwarding. Finally, we present the application of neural network techniques to improve the performance of Low-Power and Lossy Networks.

This paper organizes as follows: in Section 2, we compile a list of current related work papers on the topic of this research. Section 3 refers to the overall framework architecture. Section 4 presents an analytic model of the tower. In Section 5, we present a multi-variable fusion model based on a Feedforward Neural Network (FFNN) for estimation of structure position using an accelerometer and cable traction data as input. Finally, the discussion is closed in Section 6.

## 2. Related Work

Although several works were carried out to remotely monitor the structural health, based on IoT technology for different areas, i.e., agriculture [4], civil infrastructure [5], and smart cities [6], few works have explored IoT systems for monitoring the structural health of guyed towers. Most of these works combined monitoring the tower’s structural inclination, due to environmental parameters that can accelerate the tower deterioration, such as temperature, humidity, and wind action. Since power grids will inevitably reach remote areas, wireless ad hoc network solutions seem straightforward to achieve remote monitoring. It is possible to learn from the current literature that research works usually focus on one out of two main approaches: network and sensor technologies. Furthermore, some can also present initiatives toward machine learning techniques, which is a current trend that shows impressive contributions in a broad range of applications.

Concerning network technology, one of the first attempts for remote monitoring power lines used GPRS mobile to connect to the internet [7]. The system sends sensor data to a monitor for analysis and process. The authors in [8,9] proposed a wireless tower body monitoring system composed of accelerometers and some environmental sensors. The remote real-time monitoring of towers’ health is performed through the combination of ZigBee and GSM. The works cited above focus on measuring the structural health of towers without prioritizing the transmission and data collection strategies. Currently, most research works focus on modern approaches based on the Internet of Things and Low Power Wide Area Networks. For instance, the authors in [10] propose a tilt monitoring system considering sensing device energy consumption, link distance, and cost by adopting Low-Power Wide-Area Network (LPWAN) technology. In addition, the authors propose a LoRa multinode network connected to a real-time online monitoring system through NB-IoT. Researchers on [11] propose an online monitoring system for the tilt of power line towers, using two inclinometer sensors and Narrow Band Internet of Things (NB-IoT) network technology. In [10], the authors adopt adopts LoRa (Long Range) as a Low-Power Wide-Area Network (LPWAN) technology, as well as NB-IoT technology for a mobile data connection as a gateway to the internet. The proposed system forwards online real-time tilt information of the tower to a cloud platform through NB-IoT and LoRa multinode networks. The data history allows administrators to proceed with maintenance according to the faults pointed out by the analysis of tilt information.

Notwithstanding network technology, it is essential to present research works focusing on satellite communications. For instance, the authors in [12] propose the use of a traditional Synthetic Aperture Radar (SAR) method to obtain the tower tilt. The main challenge resides in solving the problem of layover areas distortion [13] that usually happens for line-shape objects located vertically over the ground. Thus, the authors propose removing the top-down vertical height phase by obtaining the differential phase caused by inclination displacement. In [14], based on [12], the author proposes a revised algorithm that can remove the top-bottom vertical-height interferometry phase of the tower by using the information of every tower. Then, the residual differential interferometry phase obtained after the phase unwrapping allows calculating the inclination displacement of the top position of the tower more precisely. Finally, in [15], the authors retrieve the inclination direction of the power transmission tower using geocoding. First, the backscattering coefficient separates the power line tower from the background, and a refining processing marks the tower’s position in the image. Finally, a distance-Doppler model retrieves the geographical coordinates of those marked points. The authors claim that the calculated tilt results have a 0.0025° error.

Concerning the focus on sensor technology, the research work in [16] proposes a computational algorithm based on least-squares state estimation applied to transmission line sag-tension equations through a geometric transformation to determine conductor sag levels. The estimation of the tower health considers tower tilt, conductor tension, core temperature, the inclination angle of the conductor with the horizon as input parameters. Ref. [17] also presents a similar system for online monitoring based mainly on real-time measurements, such as current through the line, conductor temperature, conductor inclination angle/sag, mechanical tension of the line, conductor oscillations. Differently, the authors in [18] focus on monitoring the transmission tower base’s vertical displacement responses, using a hydrostatic leveling system, which works as a slope sensor. The research work shows the importance of making the right choice of sensor technology.

Furthermore, the specific sensor is strongly affected by the environmental temperature, which impacts the actual evaluation of tower health. In [19], the authors propose an online monitoring technique applied to power line towers near mining, river, hillside, and other like areas. The system monitors the stress or strain on the tower that can cause foundation deformation. The stress evaluation considers sensing information on foundation settlement, inclination, and side-slip through a finite element model of the tower. The sensing technology used in the actual tower foundation is a fiber Bragg grating. The results show that the tower foundation condition is compromised when sensing data values cross the limits of the ones seen in a tower of reference that it is in good condition.

Despite previous sensor technology kinds, it is essential to present research works focusing on vibration approaches. In [20], the system generates an impulse excitation to the transmission tower to learn about the tower’s health. First, the monitoring device uses accelerometers to read the impulse excitation. Then, the method allows revealing the tower’s natural frequencies that can help evaluate the tower’s condition. Ref. [21] also presents a vibration-based transmission tower structural health monitoring system. An analyzer receives data from the acceleration sensors and calculates the tower’s natural frequencies. The method for analyzing the natural frequencies at different wind speeds is stochastic subspace identification (SSI), which can reveal the number of natural frequencies of the overall structure. The researchers test the approach by directly interfering (i.e., lifting one of the tower legs) and comparing the natural frequencies of each case. The results show that differences between before and after the interference can be the basis for judging the condition of the tower structure.

On the way to modern approaches, machine learning (ML) techniques are relevant in structural monitoring systems for fault, risk detection, and modal parameter estimation. Sensor information, such as vibration and electromagnetic response, may be input to ML models to detect faults or risk situations [22,23] and, in the case of structures with multiple potential points-of-failure, to pinpoint the location of faults [24,25]. Modal parameters, such as stiffness and position, may also be estimated through ML models [26].

In [22], the authors develop a Convolutional Neural Network (CNN) for the classification and extraction of features and detecting different damage cases in a structural monitoring system based on vibration data sampled from accelerometers. In [23], the authors apply K-Nearest Neighbors to classify different damage cases in structures having vibration data as input. The authors in [24] propose a method of detection and damage localization in structures based on vibration signals using CNN. In [25], a two-step machine learning method is proposed for the classification and location of faults in structures, using electromagnetic response data. In that method, the first step uses Random Forest to detect whether the data contain faults. Later, the processed data are the input for an artificial neural network to detect fault locations. In [26], an artificial neural network technique is developed to estimate structure stiffness after extreme events, such as earthquakes.

## 3. System Architecture

Currently, many LPWAN technologies exist, including NB-IoT, LoRa, Sigfox, and Wi-SUN, among others. When choosing a particular technology, the systems integrator should know about the unique features of the technology as well as the specific application needs and demands. In this particular case, the choice for using the Wi-SUN network is due to several reasons. First, Wi-SUN networks are usually meshed networks. This topology allows multiple, redundant connection paths for a network with communicating tower sites, unlike star-based networks. As mesh networks scale, their reliability and performance improve because the possible communication paths multiply. Second, Wi-SUN provides high data rates that are consistent throughout the network and present the lowest latency [27]. Additionally, it uses less power for listening, enabling devices to listen frequently and still endure the long-term operation [28]. Finally, Wi-SUN is an open standard, and no company exclusively owns the technology. Thus, implementing private solutions will not require black box chips or licenses with potential vendor lock-in. The standard was established in 2011 and leveraged IEEE 802.15.4g [29] and IPv6 [30] protocols to drive innovative wireless Smart Utility and Smart City applications.

The framework proposed in this research work considers the use of Low-Power and Lossy Networks (LLNs). Figure 2 illustrates an implementation of the proposed framework for monitoring the health of power line guyed towers. It consists of a set of sensors in a tower site where the information propagates through the LLN. The data hop site-to-site until the border node. The data coming from the border node can be stored in a local server or transmitted to a remote central server connected to the internet.

The overall structural design includes leaf/router nodes and a border node. Router and leaf nodes are restricted to the tower site area and include a set of standard embedded sensors (i.e., accelerometer, gyroscope, thermometer, humidity, and barometer), while specific sensors can be connected on-demand (i.e., tensiometers, weather station). Figure 3 illustrates the position of leaf, router nodes, and tension sensors in one tower site. While leaf nodes and tension sensors are on the cables, a single router node is at the top of the tower. The tension sensor connects to its respective leaf node.

The current devices operate as LLN routers, which typically run with constraints on processing power, memory, and energy (battery power). Supported traffic flows include point-to-point (between devices inside the LLN), point-to-multipoint (from a central control point to a subset of devices inside the LLN), and multipoint-to-point (from devices inside the LLN toward a central control point). On its stack proposal, IETF standardizes the IPv6 Routing Protocol for Low-Power and Lossy Networks (RPL) as the routing protocol for LLNs. RPL is a tree-based proactive routing protocol that creates acyclic graphs among the nodes to allow data exchange. The most convenient method for the current framework is the Destination-Oriented Directed Acyclic Graphs (DODAGs), one of the techniques of multipath routing over conventional single-path routing methods for implementing networks in a distributed way. The typical network arrangement to tackle the current problem is the directed acyclic graph with exactly one root node with no outgoing edges. Figure 4 illustrates a topology of the proposed solution where each tower site has four leaf nodes and one router node. The order of communication is, first, leaf nodes communicating to the router node. Next, the router node communicates to the router node of the following guyed tower site until the border node.

### 3.1. Mockup Tower

The framework included designing and manufacturing a mockup model in a reduced scale of a structure for design evaluation, testing, and demonstration purposes. Figure 5 shows a photo of the mockup tower installed inside the *LabEDin* mechanical engineering laboratory. A mockup tower in a scale factor of 1:5 was built to analyze and reproduce the dynamic responses of the actual structure. The *LabEDin* at the University of Campinas (UNICAMP) was responsible for designing and fabricating the mockup model. It is a prototype to provide at least part of the functionality of an actual guyed V-shaped tower and enables testing of the framework. The use of mockups as a tool for designing is essential for several industries. For example, the automotive device industry uses mockups to evaluate the vehicle dimensions and shapes subjected to the wind tunnel as part of the product development process.

### 3.2. Structure Design of Leaf Nodes

The leaf node comprises a MicroController Unit (MCU), a wireless communication module, a sensor module, ports to connect on-demand sensors, and a power management system. The leaf node enables sensing information of the structure and is fixed on the tower staying cables. The standard sensors of the leaf node are for movement (i.e., accelerometer and gyroscope), and environment (i.e., temperature, humidity, ambient light, atmospheric pressure). There is a tension sensor connected directly to the tower staying cable. Some sensors have a programmable threshold trigger to dispatch a wake-up signal to the MCU through the interruption ports. The sleep mode operation helps to keep a low-power operation. Furthermore, the leaf nodes can only connect to the local router node to keep the RF power consumption in a low profile. Figure 6 shows a photo of the leaf node and tension sensor attached to the laboratory mockup tower cable.

### 3.3. Structure Design of Router Nodes

The router nodes are similar to leaf nodes in the size, MCU, wireless communication module, sensor module, ports to connect on-demand sensors, and power management system. The difference is that it locates at a maximum height, according to Figure 3, which should be the most appropriate location to allow a direct line of sight to the adjacent tower site. The standard sensors of the router node are for movement (i.e., accelerometer and gyroscope) and environment (i.e., temperature, humidity, ambient light, atmospheric pressure). In addition, the router node enables specific sensing information of the structure, such as a set of sensors to monitor the weather (i.e., wind speed, wind direction, and rain gauge). The devices self-monitor their functionalities to ensure a high probability of availability. Figure 7 illustrates four router/leaf nodes and a border node being tested in a testbench.

### 3.4. Structure Design of the Border Node

The border node comprehends an MCU, a wireless communication module, an optical fiber port, and an Ethernet port. Border nodes manage the connection between the router nodes and act as an endpoint for the data from router and leaf nodes. It can also coordinate how the mesh network must behave, including more nodes and defining routing paths. Finally, it can also act as a gateway for internet access, allowing information flow to a central server. Figure 8 shows a photo of the border node presenting the interface of the monitored data.

### 3.5. Hardware Design of the System

A single board design encompasses the router and the leaf node devices to decrease the manufacturing costs and enhance scaling production. The only difference resides in the design of the border node, which has a specific purpose. The router and leaf nodes are equipped with a long-term power supply and are design to operate based on strict energy-saving methods. In this way, it is possible to keep all the necessary electronic components, using just a single form factor board. Whenever there is a need for fabricating a router or leaf node, the assembler needs to populate the standard board only with the necessary electronic components. It also favors packaging protection when we define a hefty standard to protect against dust and electromagnetic interference. Next, we describe in detail each element that composes the nodes.

#### 3.5.1. MCU

The master chip used in this monitoring system is an RX65N. It has the advantages of low power consumption, high performance, and low cost. The operation frequency is 120 MHz, 32 bit core, on-chip floating-point unit, up to 2 MB flash memory (supportive of the dual bank function), 640 KB SRAM, and various communications interfaces, including Ethernet MAC, quad SPI, CAN, 12 bit A/D converter, RTC, CMOS camera interface, Graphic-LCD controller, and a 2D drawing engine. These are standard computing power needs that allow a node to accomplish the remote monitoring function, including pre-processing and data handling to the multihop network.

#### 3.5.2. Wireless Communication

The network needs to deliver data traffic efficiently at a low power profile. Then, it is acceptable to work with low speeds and a reasonable degree of data loss. IEEE 802.15.4g is a physical layer created by the IEEE 802.15.4g Smart Utility Networks Task Group (Wi-SUN) to provide a standard for the framework described in the current investigation. Usually, it facilitates the deployment of enormous scale process control applications, such as the utility smart grid network capable of supporting geographically large networks with minimal infrastructure. In this case, the best-chosen candidate is the RAA604S00 wireless communications IC with built-in RF peripheral components and hardware supporting IEEE 802.15.4 g/e. The RF frequency range covers from 863 up to 928 MHz; the modulation is 2FSK/GFSK with a data rate of up to 300 kbps. It is among the leading LPWAN communication technologies and the industry’s lowest level of power consumption.

#### 3.5.3. Sensors

Cable tension sensor: The cable tension sensor is important because it is a straightforward way to verify whether the cable is loose or over-tight. The sensing device is a load cell to monitor the safety of cranes, rolling bridges, trolleys, and elevators. It is installed directly on the cable using screws and bolts without the need for cutting. The principle of operation consists of measuring the angle deviation over a “V” shape formed by the cable path on the mechanical structure of the cell. Thus, this method preserves the integrity of the cable because no cuts or exceeding deflections are necessary to achieve the cable tension measurement. The electronic side of the sensor relies on a Wheatstone bridge of 350 Ohms, and the load limit measurement ranges from 10.000 a 57.000 Kgf. Figure 9a shows the diagram of the sensor connected to the cable, and Figure 9b shows the actual tension sensor attached to the tower cable.

Accelerometer and gyroscope: Measurement of displacement or configuration is essential for a guyed tower, particularly for a long-span structure, and there is no single perfect technology for the purpose. Nevertheless, an accelerometer is the most straightforward measure technology that converts mechanical motion into an electrical signal. Thus, it is an electromechanical device that measures acceleration forces, whether caused by gravity or motion. Concerning the gyroscope, the main difference is that it senses rotation, which is essential when strong winds cause rotational forces. The MPU6050 module is a three-axis acceleration sensor. It has angular velocity sensing ranges at ±250, ±500, ±1000, and ±2000°/s (dps). The accelerometer ranges are ±2 g, ±4 g, ±8 g and ±16 g. The MCU can communicate with the module through IIC interface pins, SCL and SDA.

Temperature and humidity: The atmospheric corrosion of metallic materials causes tremendous economic losses every year worldwide. Thus, it is meaningful to predict corrosion loss in different field environments. Generally, the corrosion prediction method includes feeding a model of the corrosive environment with history data collected from long-term temperature and humidity variables. Then, temperature and humidity sensors are important in the task of predicting the guyed tower health. The HDC2010 is an integrated humidity and temperature sensor with very low power consumption. The motivation of choice is that the sensing element is on the bottom part of the device, making it more robust against dirt, dust, and other environmental contaminants. A second important feature is the heating element to dissipate condensation and moisture. The device supports a 1.8 V power supply voltage combined with programmable sampling intervals that leverage online time for battery-operated systems. The HDC2010 is factory-calibrated to 0.2 °C temperature accuracy and 2% relative humidity accuracy and includes a heating element to burn away condensation and moisture for increased reliability. The HDC2010 supports operation from −40 °C to 125 °C and from 0% to 100% relative humidity.

Ambient light: Depending on the amount of sunlight, the guyed tower can increase in size. It is a natural physical phenomenon called thermal expansion, where heat causes an increase in volume, making metallic structures increase by a few centimeters. That is why the amount of light must be monitored because it also contributes to the tower’s tilt. In this way, the OPT3001 is the light sensor device included in the current framework. Furthermore, the light information of the region can help monitor the battery status, not letting the device die out of energy charge. The sensor’s spectral response tightly matches the photopic response of the human eye and includes significant infrared rejection. The device’s measurement range is from 0.01 up to 83 k lux, without selecting intermediate scales manually. This capability allows light measurement over a 23 bit effective dynamic range. Measurements can be either continuous or single-shot. The control and interrupt system features autonomous operation, allowing the processor to sleep while the sensor searches for appropriate wake-up events to report via the interrupt pin. The sensor reports data over an I2C- and SMBus-compatible, two-wire serial interface. The low power consumption and low power-supply voltage capability of the OPT3001 enhance the battery life of battery-powered systems.

Weather station: Although general weather information is broad, it is primarily local information. For example, patterns and cycles in the atmosphere impact the oceans, the patterns and cycles of the ocean affect the atmosphere, the larger-scale features affect the smaller-scale features, and the smaller-scale features can affect the large-scale features. Then, the local conditions are somehow random and can uniquely affect the tower structure. That is why the logging of local weather is vital for data analysis. According to Figure 10, the weather station consists of three sensing elements: (a) rain gauge, (b) wind speed, and (c) wind direction. Figure 10d shows a photo of the sensors installed in the tower structure. The choice of the sensing elements for this framework is irrelevant since there is a variety of them. However, the designer must observe the most appropriate equipment for the considered environment, such as robustness and long-term lifetime. For example, the most popular anemometers are the cup with a wind vane, the sonic, and the propeller. For a rain gauge, the principle upon which it works is the method of the tipping bucket principle. Since the interface of these devices is simple, it is easy to connect different sensing elements to the on-demand port of the leaf/router nodes.

#### 3.5.4. Energy Management

The power supply solution is a design that makes the system power self-sufficient, and the method of energy management is different for router and leaf nodes. The leaf node operates with battery only and is programmed to make strict use of energy. On the other hand, the router node uses an energy controller, solar panels, and lithium batteries. In this case, the solar controller derives energy to charge the lithium batteries while powering the overall system. The chosen component for this task is the BQ25570 device that extracts power in the range of microwatts (μW) to milliwatts (mW). It is used with high output impedance DC sources, such as photovoltaic (solar) or thermal electric generators (TEG) that enable the circuit to better handle the overvoltage. In addition, the battery management features ensure that a rechargeable battery is not overcharged or depleted beyond the safe limits of the battery and system operation.

### 3.6. Software Design of the System

The software design of the system includes the heuristics of leaf/router and border nodes. The most notable feature of the software is that the algorithm design can mainly focus on sensor processing. The network signaling is strictly treated by the Wi-SUN protocol stack, making communication very transparent from the perspective of the designed algorithms. The transparent communication process is possible, due to the Wi-SUN FAN self-forming and self-healing features. It means that adding new devices to a network is easy, and it automatically re-routes to the gateways when a pathway fails. Thus, the destination and the forwarding of the messages are within the Wi-SUN’s protocol functionality, and it is outside of the software scope to handle network signaling.

The main difference between leaf and router nodes is that the router keeps a “passive” state by handling the messages of other tower sites flowing through, while the site leaf node’s devices can go dormant during the timeout period. Next, we describe the internal process of the program and the initialization of each part of the system.

#### 3.6.1. Heuristic Description for Leaf and Router Nodes

The heuristic of the node depends on its designation. A single node designated as leaf features short-range communication defined by its function and energy requirements. The standard function of the leaf node is to collect information coming from the sensors and deliver the data to the router node and go to sleep. The operation of the leaf node focuses on energy saving by following the steps of waking up, collecting data from sensors, pre-processing, packing data, transmitting information, and sleeping. These sets of actions ensure low power consumption. Figure 11 illustrates a flowchart of the sequence of procedures performed by the leaf node.

A single node designated as a router features long-range communication defined by its function and energy requirements. In this case, the router does not sleep and consumes more power to keep active links to adjacent tower sites. Most of the time, the router node forwards data from the previous tower router node to the next tower router node. The ultimate destination of the collected sensor information is the central server behind the border node gateway.

The main idea of the current framework design considers the nodes’ arrangement in a mesh network topology operating in a multi-hop packet routing, according to the topology seen in Figure 4. Then, the router’s primary function is to concentrate the collected sensor data from leaf nodes and transfer information to the next tower’s router node, having the border node at the sub-station as an end point. In this case, it is better to understand a router node having a dual function. First, it acts by passing information from other nodes and, second, passing its information by working as a leaf node. Thus, in router node mode, it waits for leaf nodes to send data from sensors. Then, it arranges data from leaf nodes, determines the destination address, and sends all leaf nodes’ data to the router node of the next tower. Furthermore, when the node acts as a leaf, it aggregates the received data and the information from the current site leaf nodes and passes it forward.

#### 3.6.2. Border Node and Central Server

From the perspective of the network, the border node acts as the primary coordinator of the mesh network by handling all join requests and validating and authenticating all nodes. Thus, nodes can communicate with each other by using IPv6 addresses, and the Wi-SUN network deals with the routing problem. A node starts participating in the network after the border node accepts its initial handshaking requests and assigns an IPv6 address whenever they “turn on”. Next, the node learns which other nodes are direct neighbors who can send messages. When a new node completes the initial handshaking process, the routing table of the border node updates. Additionally, the assigned address of the nodes is associated with the geographic localization. By standard, a node has an IP address, MAC address, and geographic localization of the device that occurs at the network association process. The nodes learn the IPv6 address of the border node, which is the default destination of the leaf sensing information. The border node controls the routing tables, referencing the router nodes to know the best paths for message passing. The border node also keeps a small record of recent messages it receives. If the connection to the server is lost, the border continues to store events locally and syncs to the server when the network link is re-established.

All nodes belonging to a specific power path send their sensor’s data to the border node located at a substation. Whenever a message is received, the border translates the IP from the received message to an identification tied to a specific guyed tower site. A second function of the border device is to allow real-time verification of the sensors’ state through a human–machine interface (HMI). The HMI aids the technical staff in the need for in-site visits according to the tower site information. Figure 12 illustrates the software interface of the border node HMI. The screen is touch-sensitive, and the user can choose the tower site by touching the sides of the screen.

The message’s contents are validated and re-arranged to be sent forward to the central server, using a secured connection. Currently, data cross the internet using an intermediate server. Over the internet, the established links use SSL encryption for further protection of the messages exchanged. We used MQTT as a messaging protocol for the Internet of Things (IoT) for the current framework. MQTT is a significantly lightweight publish/subscribe messaging transport ideal for connecting remote devices with a small code footprint and minimal network bandwidth.

The primary function of the central server is to receive data, pre-process them, store them in the database, and display them in a meaningful manner. Figure 13 illustrates the software interface that the user can access. The agent interested in monitoring the condition of any tower site can analyze the data stored in the central server. In this way, it can handle the computer interface depicted in Figure 13a to set access details of any tower site with sensing information for each cable. Furthermore, the history data can be seen according to what Figure 13a–d describes. Functions, such as setting alarms whenever there is a threshold crossing, are possible. The data at the server serves to accommodate multiple users to access stored data and valuable functions. That idea allows integration, remote visibility of the tower site, sending out alarms, and having prediction systems running simultaneously.

## 4. Analytical Model of the Tower

The current framework includes a complete set of sensors to make the solution all-purpose. Consequently, the designer can follow the current hardware guidelines in this research work but limit the used sensors for its algorithmic approach. Furthermore, to demonstrate modern approaches, we consider a simplified model based on vectorial analysis that helps us better understand the tower’s behavior and ML techniques. While it is impossible to capture the flow details and other perturbations around the actual structure within this method, the overall displacement and forces are well reproduced, reflecting practical information captured by the sensors. Furthermore, synthetic data are helpful for the ML approach since they are practically possible to create unlimited data for ML training. Figure 14 illustrates a simplified analytical model of the mockup tower. The V-shaped guyed tower simplifies to a T-shaped model where cables attach at the top side edges. There are four monitored cables. Each cable has a tension sensor to read the forces and an accelerometer to read the gravity vector information. The forces, denoted in the plot by the spring symbol over the cable, follow Hooke’s law, where the force needed to extend or compress a spring by some distance scales linearly to that distance. Thus, the tension force $F_{cb}$ over a cable is as follows:(1)$$F_{cb}=-kx$$
where *k* is a constant factor characteristic of the cable, and *x* is the total deformation of the cable concerning the original length of the cable. On the other hand, the quantity of interest from the accelerometer is the gravity vector g. The vector g is an essential reference because it is always pointing to the same vertical direction when the device is static. In this way, it is possible to calculate the angle $\theta_{cb3}$ of the direction of the Cable 3 and the vector g. In addition, notice from Figure 14 the explicit correlation between tension, given by Equation (Equation 1) and the angle $\theta_{cbx}$ of each cable.

The accelerometer data must receive careful analysis when dealing with global and local axes reference. The supplied data of the accelerometer device are considered a local axis reference. Thus, the local axis reference must be converted to a global axis. Notice in Figure 15 that the *y*-axis local reference of the devices is always parallel to the cables and rotated concerning the global axis represented in the left corner of the plot. If the accelerometer data are imported in the global reference, the local axes coincide with the global axes, and the g vector points to an arbitrary direction as is shown in Figure 15a. Thus, since we know that g vector points to the vertical direction, it is possible to calculate a rotation matrix and apply an operation to achieve the correct reference. Figure 15b illustrates the correct reference when we apply rotation on the local coordinates of the accelerometer.

The most important information from the accelerometer is to obtain the angle between the *y*-axis and the g. Furthermore, the most critical requirement is to ensure that at least one axis is parallel to the cable. Since, in this case, the *y*-axis is parallel, then the *x*- and *z*-axes relative to the global reference are irrelevant. The angle θ between the *y*-axis and the g is given as follows:(2)$$\theta =\mathrm{arc}\cos\left(\frac{v\cdot u}{\left||v||\cdot ||u|\right|}\right)$$
where v represents the considered base vector parallel to the cable and v represents the vector g. Thus, we define $\theta_{cbx}$, according to Equation (Equation 2), as the angle between the *x*th-cable and the vector g.

The analytical model allows us to know g vectors and forces of each cable as a function of the tower inclination. Figure 16 illustrates a diagram showing the proposed simulation of the tower analytical model. Thus, we generate data by setting a displacement from the tower model’s initial x and y center position according to a random variable. The chosen random variable is Gaussian distributed with zero mean and unitary variance. The gray dots represent the displacement around the initial position of the tower. Notice that as the top center of the tower changes from the initial position, the cable forces on positions 1, 2, 3, and 4 change accordingly to the promoted displacement of the tower center. The data of the forces, accelerometer axes, and g relative to each cable are recorded to be used for analysis and in the machine learning training process. A MATLAB script calculates all related variables and stores them in a file that feeds the neural network for training. The tower’s *x* and *y* displacements follow a Gaussian random variable of zero mean and unitary variance.

## 5. Evaluation of the Tower Parameters for Machine Learning Estimation Model

The constant monitoring and identification of damage risk in a transmission line tower are essential for long-term health. At present, the traditional methods to identify risks are to assess the features in a physical tower under harsh testing loads to carry out the reasons for likely damage. However, since these methods are expensive, it is economical and feasible to use long-term health monitoring data and more straightforward methods to identify tower damage risks directly. In addition, machine learning is an attractive method to analyze data and learn from it, making decisions and forecasts for events in the real world. An extra advantage comes when the fusion of input parameters can cover features of the natural phenomenon that simplified analytical methods fail to obtain.

Among many machine learning algorithms, the developed neural network receives multiple correlated inputs for estimation of another variable of the system, a vital algorithm of estimation modeling machine learning. It is often used to solve classification and regression problems, and the learning and prediction speeds are good. The proposed method uses the multiple input variable fusion method, the transmission line tower displacement, and cable forces to improve the estimation accuracy. The variables collected from the system, such as acceleration, angular velocity, cable tension, temperature, atmospheric pressure, wind direction, and wind speed, are indicatives of changes in tower structural parameters. These parameters are necessary to detect structural failures (i.e., loosening of the stay cables, displacement, and vibrations) that can cause catastrophic events, such as tower collapse or even cable rupture.

### 5.1. Data and Variables Correlation

The monitoring system for cable-stayed towers comprises several sensors that monitor the state of the structure. The system model estimates traction across four cables and angle between the cable direction and g vector through multiple three-axis accelerometers. These variables have a cause-and-effect relationship to each other. One way to measure and understand the relationship between data is the correlation coefficient. A higher absolute correlation coefficient implies a more significant relationship between the data.

Before using the data for training, it is interesting to evaluate what correlation features are behind the collected samples. It is essential to inspect the correlation among variables to save computing resources by eliminating the irrelevant ones from the training phase. For the current analysis, we use the Pearson correlation coefficients to measure the linear correlation between two data sets. The Pearson coefficient ρ is calculated as follows:(3)$$\rho =\frac{\sum_{i=1}^{n}\left(x_{i}-\bar{x}\right)\times \left(y_{i}-\bar{y}\right)}{\sqrt{\sum_{i=1}^{n}{\left(x_{i}-\bar{x}\right)}^{2}}\times \sqrt{\sum_{i=1}^{n}{\left(y_{i}-\bar{y}\right)}^{2}}}$$
where $x_{i}$, $y_{i}$ are the samples, $\bar{x}$, $\bar{y}$ are the average value and *n* is the total amount of samples. The current results comes from a simulation where we consider n=10,000. Notice that the product of sample standard deviations divides the covariance, and the result is a normalized measurement where −1<ρ<1. Table 1 shows a classification range for the values of ρ, where it allows us to consider if the quantity is irrelevant for the machine learning training phase. Usually, quantities below the weak range are discarded.

The current analysis of ρ considers the data generated from the simulation of the analytical model from previous section. It is vital to remind that the considered quantities are the tension, accelerometer axis, and g vectors. The technique of drawing out information from multiple quantities and using it for estimating is called sensor fusion [31,32]. Table 2 show the data calculated from the data set using Equation (Equation 3). There, X and Y are the displacement coordinates of the tip of the tower; T1, T2, T3, and T4 correspond to the tension of the cables; and A1, A2, A3, and A4 correspond to the angle between the *y*-axis of the accelerometer and g vector.

The relation between the quantities is also closely investigated. Figure 17 shows a plot of the correlation between (a) A1 and A4; (b) A2 and A3. Notice the inverse relationship at Figure 17a,b that reflects the stretching and loosening of the cables as the model of the tower displace. Figure 18 shows a plot of the correlation between (a) T1 and T3; and (b) T2 and T4. Notice the direct relationship at Figure 18a,b that reflects the stretching and loosening, also validating the model.

### 5.2. The Machine Learning Predictive Model

A Feedforward Neural Network (FFNN) was chosen to solve the regression problem, which is to estimate the position of the tower displacement or cable tension in an analytical model designed for the initial experiments of the structural monitoring system. The FFNN contains all neurons in a layer directly connected to all neurons in the next layer. The information flow starts at the input layer and goes in the same direction until the output layer. The neural network architecture is composed of the input layer, the next layers are two hidden layers and the output layer. Figure 19 shows the neural network architecture used in the experiments. The number of neurons defined in the input layer is the number of features used to estimate the parameters. The hidden layers have 64 neurons each, and the output layer has 2 neurons for X Y estimation and 1 neuron for cable traction. The hidden layer activation function is the Rectified Linear Unit (*ReLU*).

The data used in the FFNN training process were 10,000 records obtained from the analytical model. The features considered in model were the position, angles information, cable traction, but the model can be retained for considered new features of monitoring system, e.g., temperature and wind information. The features with weak to very strong correlation coefficients were chosen for the estimation of the parameters because they present a strong linear relationship between the features, facilitating the adjustment of the model in the training process. However, in some experiments, some features with smaller correlation coefficients were inserted for evaluating the data fusion with moderate to very strong correlation.

The parameter estimation is considered a regression problem where we want to find the tension of the cables and the displacement of the X and Y axes from the initial position. We used a combination of features obtained from the simulations of the analytical model from the previous section. From the total amount of data samples, 80% are for neural network training, while the remaining 20% are for testing the estimation.

Before using data for training the neural network, it is vital to scale the samples to change the range of the values. This is because many machine learning algorithms perform better or converge faster when the features are relatively similar. For the current investigation, we used MinMaxScaler, which preserves the shape of the original distribution and is calculated as follows
(4)$$x_{norm}=\left(x-x_{min}\right)/\left(x_{max}-x_{min}\right)$$
where $x_{norm}$ is the normalized sample, *x* is the considered sample, and $x_{min}$ and $x_{max}$ are the minimum and maximum values of the data set.

The training model process used the Mean Absolute Error (MAE) loss function, which is minimized during weight adjustment. The optimizing algorithm, the method used to train the model, was the Adam with a learning rate of 0.01. The metric of evaluation of the training process was accuracy, MAE, and Mean Square Error (MSE). The epochs number of training was 1000, and 20% of the data was used to validate the neural network model in the training process.

### 5.3. Results for the Estimated Quantities of Interest

At the current moment of the investigation, it is a problem to face the shortage of data for training deep learning models. It means that the framework is ready to provide data, but it takes time until there are enough data for training the neural network. Henceforth, we deal with cases in which data sources are scarce, and thus the simulation of data becomes a viable option. This shortage is a problem since the accuracy of a model is highly dependent on the size of the dataset when working with self-learning artificial intelligence. It is expected that other fields in machine learning study face a shortage of data. This can prevent researchers from obtaining conclusive results.

A solution to the data shortage is to use synthetic training data when training a deep learning neural network. By using synthetic data, both the problem of data shortage and the labor intensity is reduced. Firstly, the data shortage problem will be eradicated since the data available through synthetic data are theoretically infinite. By having a theoretically infinite dataset, the researcher can simulate a dataset as large as they need it to be.

Evaluating the machine learning algorithm is an essential part of successfully estimating the quantities of interest. The current model may yield satisfying results when evaluated using a metric, i.e., accuracy score, but may give poor results when evaluated against other metrics, such as logarithmic loss. For the current investigation, the most appropriate metrics are Normalized Root Mean Square Error (NRMSE), Mean Absolute Percentage Error (MAPE), and determination coefficient (R^2^). The NRMSE is a metric that indicates normalized error between 0 and 1. The MAPE is a metric that measures the percentage of error. The R^2^ varies between 0 and 1 and expresses the variance of the estimated data explained for the linear model. The respective formulation for each metric is as follows:(5)$$NRMSE\left(y,\hat{y}\right)=\frac{\sqrt{\frac{1}{n_{samples}}\sum_{i=0}^{n_{samples}-1}{\left(y_{i}-{\hat{y}}_{i}\right)}^{2}}}{std(\hat{y})}$$
(6)$$MAPE\left(y,\hat{y}\right)=\frac{1}{n_{samples}}\sum_{i=0}^{n_{samples}-1}\frac{|y_{i}-{\hat{y}}_{i}|}{max(\epsilon ,|y_{i}|)}*100$$
(7)$$R^{2}\left(y,\hat{y}\right)=1-\frac{\sum_{i=0}^{n_{samples}-1}{\left(y_{i}-{\hat{y}}_{i}\right)}^{2}}{\sum_{i=0}^{n_{samples}-1}{\left(y_{i}-\bar{y}\right)}^{2}}$$
where *y* represents the actual data, $\hat{y}$ the estimated data, $\bar{y}$ is the mean, $n_{samples}$ is the number of samples, std($\hat{y}$) is population standard deviation of estimated data, and ϵ is small positive number utilized to avoid an error result when *y* is zero.

Table 3 presents the tension estimation results for some attempts choices of available features. Features are not easy to choose, and the results reveal that a method of choice beyond correlation is required. Additionally, the results highlight that the metric analysis is a good choice for choosing suitable features for successful quantities of interest estimation.

Table 4 presents the X and Y displacement estimation results for some attempts choices of available features. Again, the results highlight that the metric analysis is a good choice for choosing suitable features. In this particular case, the use of raw features X1, Y1, Z1, X2, Y2, *…*, Z4 seems to produce much better results. It is possible that the neural network can perceive details from X1, Y1, Z1, X2, Y2, *…*, Z4, which are not present in A1, A2, A3 and A4.

## 6. Conclusions

We have seen so far that the collapse of overhead power line guyed towers is one of the leading causes of power grid failures, subjecting electricity companies to pay considerable, high-value fines. Additionally, city blackouts resulting from the collapse of cable-stayed structures cause financial losses of millions of dollars for industries and threaten sensitive sectors, such as hospitals that depend on an uninterrupted energy supply.

We pointed out that traditional solutions, such as ground inspection by foot, seem ineffective since in-site periodic inspections are sometimes insufficient to detect hidden defects in such structures. Furthermore, the interval between visits and expensive personnel allocation makes monitoring/predicting defects difficult. Other monitoring methods, such as wired communication and video monitoring, are costly and hard to scale.

Henceforth, we proposed a novel and complete framework for remotely monitoring mechanical stresses and deformation of guyed towers. The proposed framework considers wireless monitoring of environmental threats in two ways. First, the analysis of remotely recorded data allows finding evidence of risk in the guyed tower structure, due to environment-critical events and, subsequently, a correlation between the guyed tower structural failure and harsh environment degrading factors. Second, the constant online sampling of data in the cable traction allows real-time monitoring of the guyed tower health, which helps to prompt personnel to act immediately.

Furthermore, the reduction of in-field inspections and the adoption of ubiquitous monitoring increase the productivity of the asset management wellness and ensure the quality of service for the consumer in the power transmission segment of the electric sector. Therefore, we expect the risk of collapse of guyed cable towers to reduce when the proposed framework is applied. In addition, we hope to facilitate inspection and optimize maintenance since remote monitoring allows to switch from periodic to on-demand maintenance visits. We also showed that it is possible to estimate tower parameters using a neural network and sensor fusion from multiple sensors. The proposed method uses multiple input variable fusion, such as accelerometers and tension sensors, to estimate the tower’s displacement. These estimations help address the structural health of the tower against failures (i.e., loosening of the stay cables, displacement, and vibrations) that can cause catastrophic events, such as tower collapse or even cable rupture.

Finally, we hope to facilitate inspections, increasing the productivity and quality in asset management since remote monitoring allows to switch from periodic to on-demand maintenance visits.

## Figures and Tables

**Figure 1 sensors-21-06173-f001:**
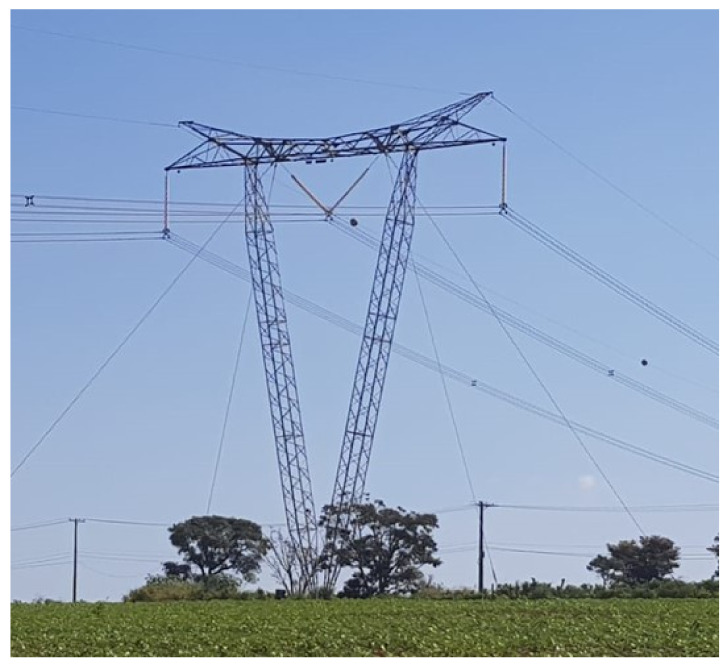
Typical guyed-V steel lattice tower commonly seen in the Brazilian power grid.

**Figure 2 sensors-21-06173-f002:**
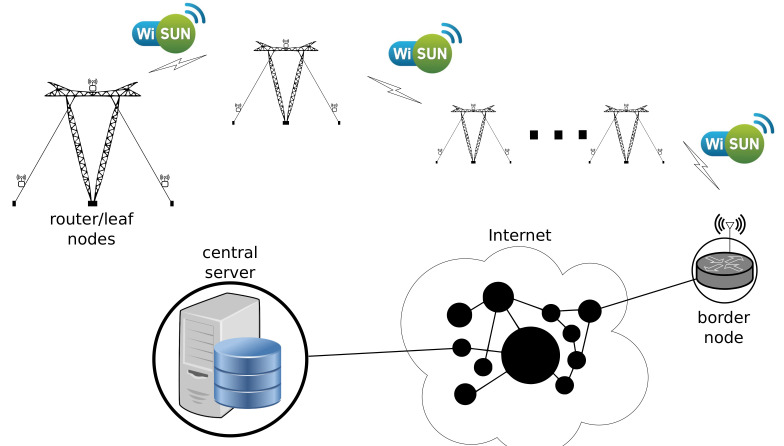
The proposed solution framework.

**Figure 3 sensors-21-06173-f003:**
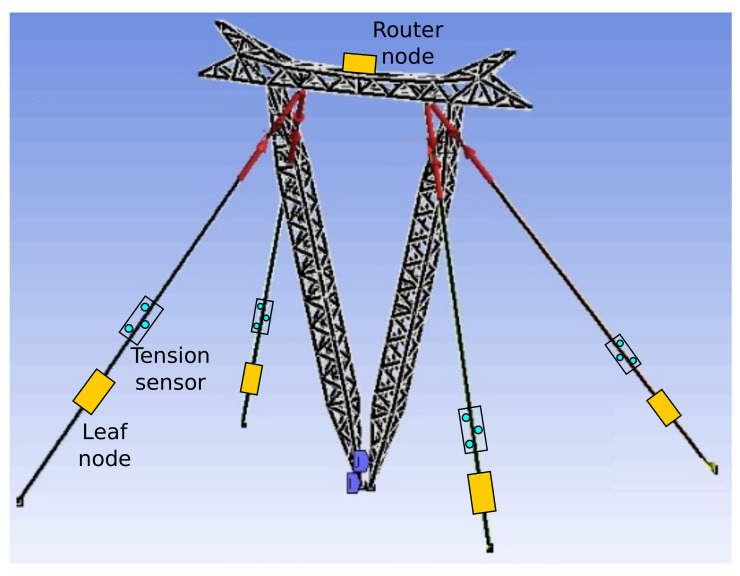
Virtual model of an guyed-V steel lattice tower and the placement of the devices and sensors.

**Figure 4 sensors-21-06173-f004:**
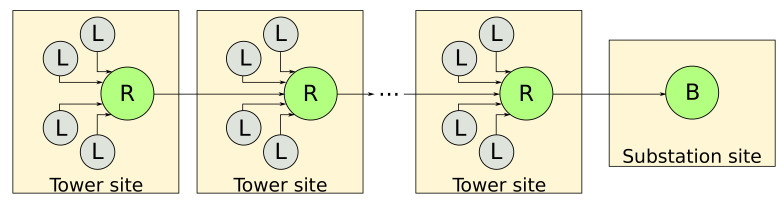
The node arrangement in the context of multipath routing.

**Figure 5 sensors-21-06173-f005:**
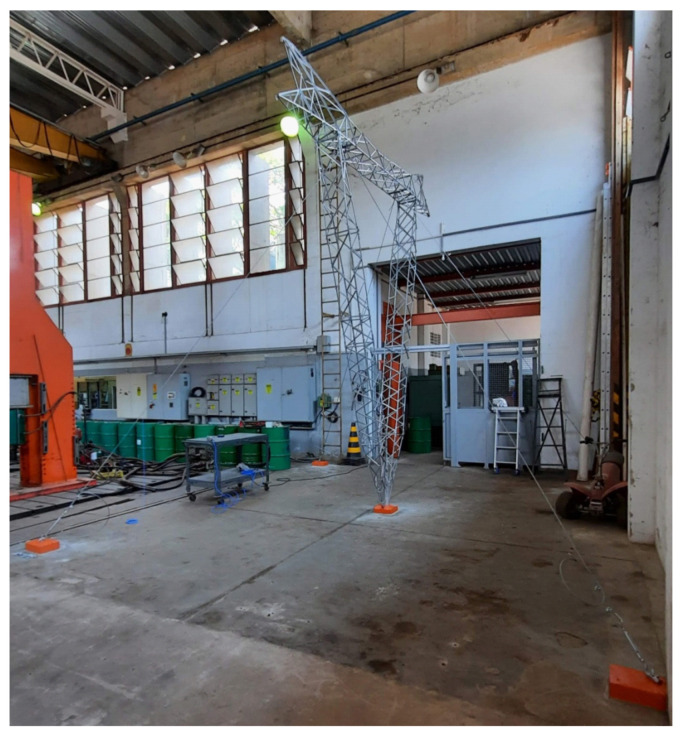
A photo of the mockup tower installed inside the *LabEDin* mechanical engineering laboratory.

**Figure 6 sensors-21-06173-f006:**
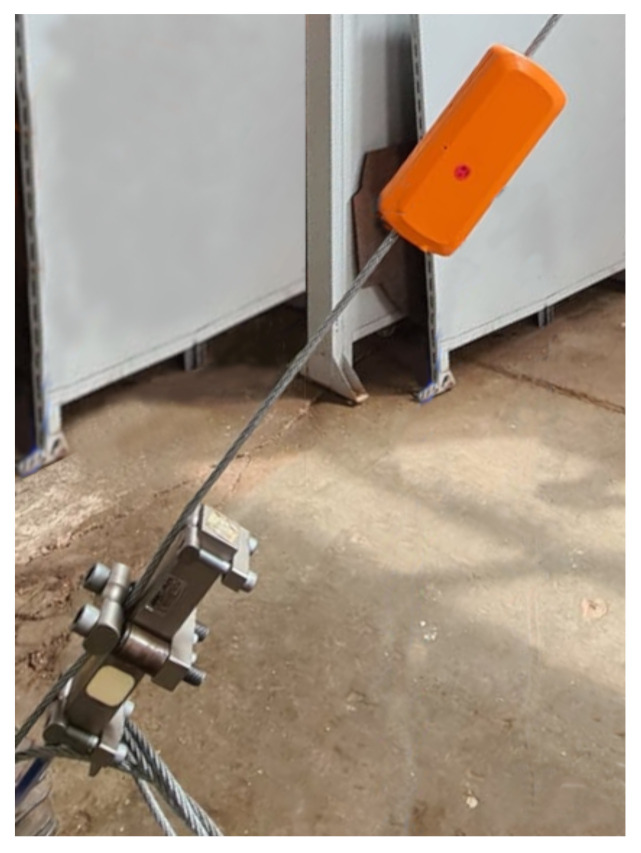
Photo of the leaf node and tension sensor attached to the tower cable.

**Figure 7 sensors-21-06173-f007:**
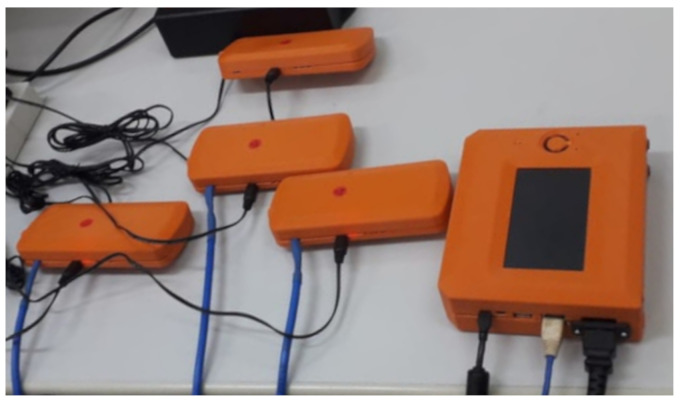
A photo of four router/leaf nodes and a border node being tested.

**Figure 8 sensors-21-06173-f008:**
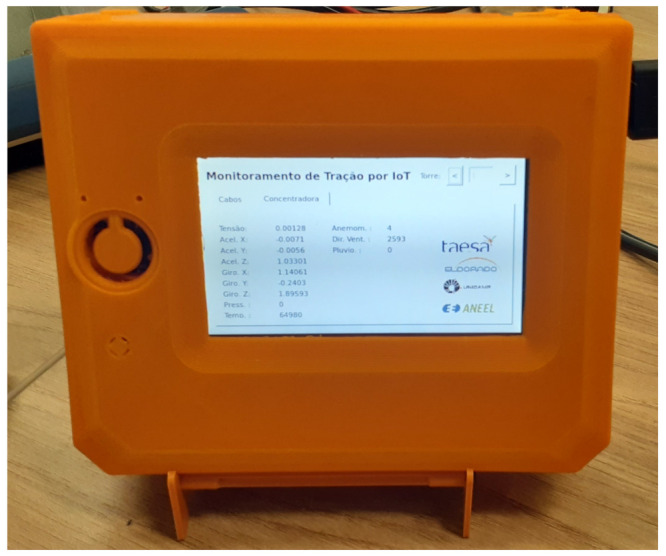
Photo of the border node presenting the interface of the monitored data.

**Figure 9 sensors-21-06173-f009:**
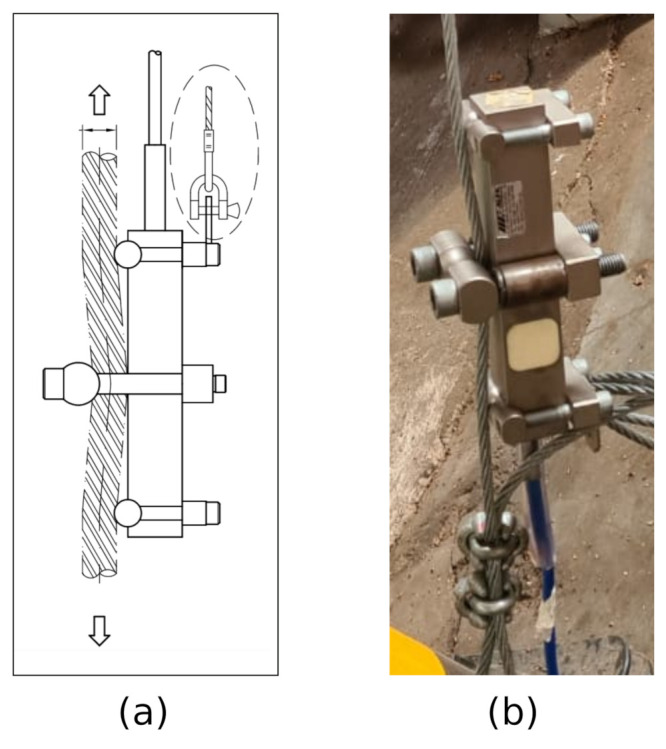
The tension sensor: (**a**) diagram and (**b**) actual photo.

**Figure 10 sensors-21-06173-f010:**
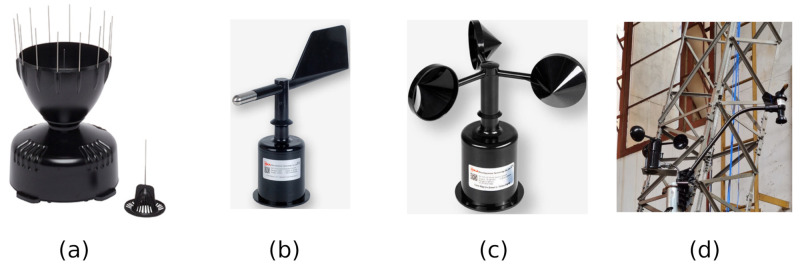
The weather station sensing elements: (**a**) rain gauge, (**b**) wind direction, (**c**) wind speed, and installed sensing elements on the mockup tower, and (**d**) all sensors installed in the tower.

**Figure 11 sensors-21-06173-f011:**
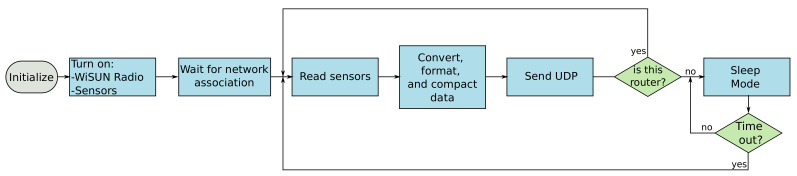
A generic flow chart for leaf and router nodes.

**Figure 12 sensors-21-06173-f012:**
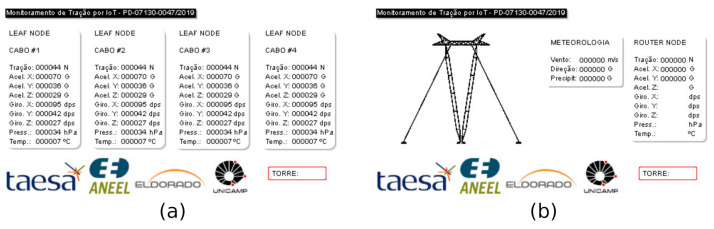
User interface samples from the border node HMI: (**a**) real-time information from leaf nodes; (**b**) real-time information from the router node and weather station.

**Figure 13 sensors-21-06173-f013:**
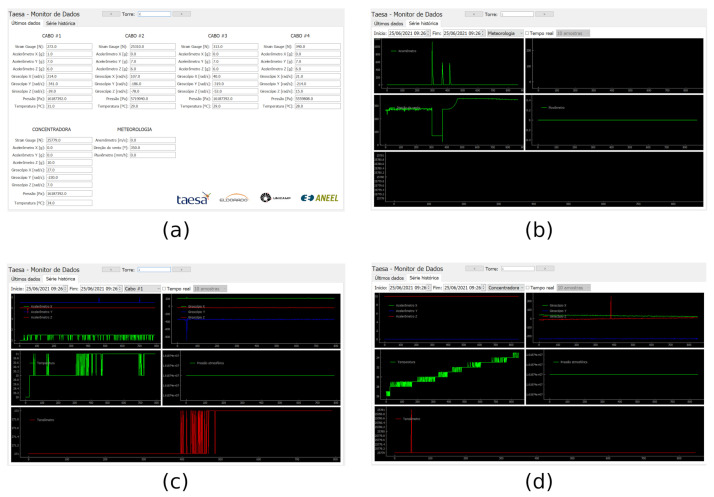
User interface samples from the central server: (**a**) View of leaf nodes data from sensor cables, accelerometers, and weather stations, (**b**) View of history data plot of the weather station, (**c**) View of history data plot of cable 1, and (**d**) View of history data plot of the central accelerometer.

**Figure 14 sensors-21-06173-f014:**
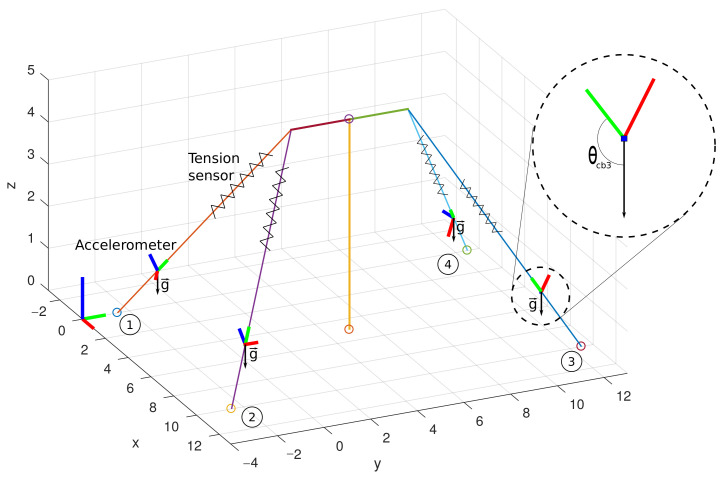
Analytical representation of the mockup tower.

**Figure 15 sensors-21-06173-f015:**
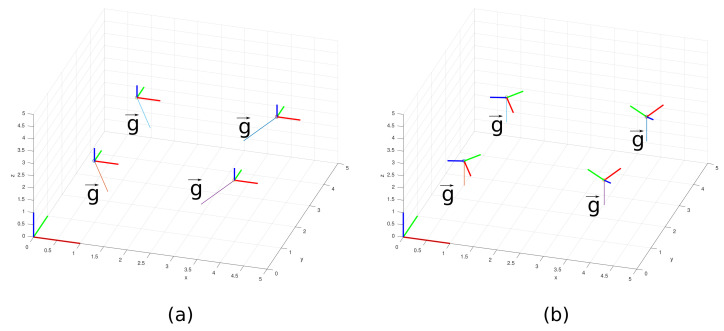
Axes references for (**a**) local and (**b**) global.

**Figure 16 sensors-21-06173-f016:**
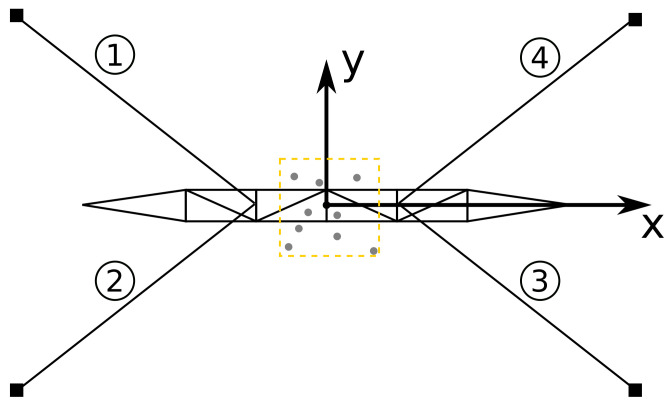
Tower displacement simulation diagram. The tags 1, 2, 3, and 4 shows the location of the tensor and accelerometer sensors of each cable.

**Figure 17 sensors-21-06173-f017:**
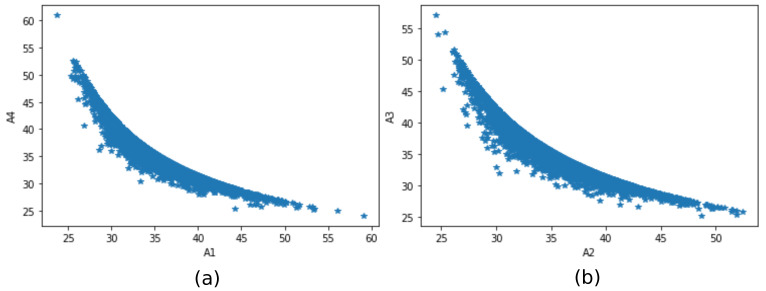
Relationship between (**a**) A1 and A4, (**b**) A2 and A3.

**Figure 18 sensors-21-06173-f018:**
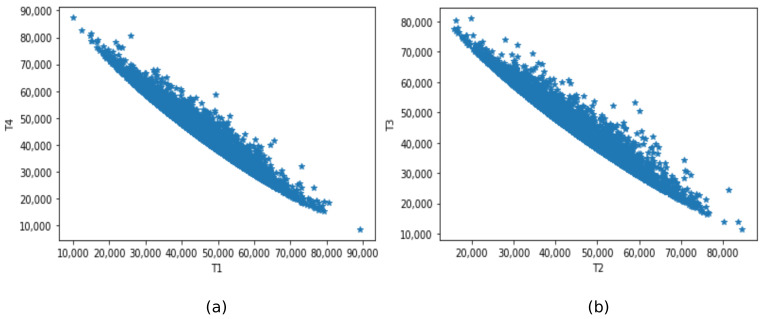
Relationship between (**a**) T1 and T4, (**b**) T2 and T3.

**Figure 19 sensors-21-06173-f019:**
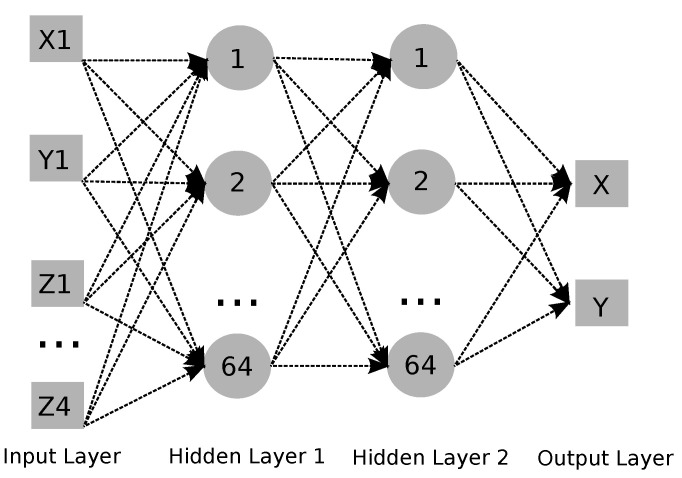
Neural network architecture.

**Table 1 sensors-21-06173-t001:** Classification ranges for the very values of ρ.

ρ	Correlation
±0.9	Very strong
±0.7 up to ±0.9	Strong
±0.5 up to ±0.7	Moderate
±0.3 up to ±0.5	Weak
0 up to ±0.3	Irrelevant

**Table 2 sensors-21-06173-t002:** Correlation between dataset variables.

	X	Y	T1	T2	T3	T4	A1	A2	A3	A4
**X**	1.000	0.002	0.703	−0.701	0.702	−0.704	−0.697	0.696	−0.696	0.698
**Y**	0.002	1.000	0.705	0.703	−0.702	−0.704	−0.699	−0.698	0.697	0.698
**T1**	0.703	0.705	1.000	0.010	0.005	−0.978	−0.986	0.006	0.011	0.976
**T2**	−0.701	0.703	0.010	1.000	−0.977	0.005	0.006	−0.987	0.975	0.012
**T3**	0.702	−0.702	0.005	−0.977	1.000	0.007	0.012	0.975	−0.986	0.009
**T4**	−0.704	−0.704	−0.978	0.005	0.007	1.000	0.976	0.010	0.008	−0.986
**A1**	−0.697	−0.699	−0.986	0.006	0.012	0.976	1.000	−0.022	−0.028	−0.947
**A2**	0.696	−0.698	0.006	−0.987	0.975	0.010	−0.022	1.000	−0.948	−0.027
**A3**	−0.696	0.697	0.011	0.975	−0.986	0.008	−0.028	−0.948	1.000	−0.024
**A4**	0.698	0.698	0.976	0.012	0.009	−0.986	−0.947	−0.027	−0.024	1.000

**Table 3 sensors-21-06173-t003:** Tension estimation results for some attempts choices of features.

Estimation	Features	NRMSE	MAPE	R^2^
T1	A1, A2, A3, A4	0.0755	1.4559	0.9943
T2	A1, A2, A3, A4	0.1057	1.7218	0.9888
T3	A1, A2, A3, A4	0.0596	1.0897	0.9965
T4	A1, A2, A3, A4	0.1234	2.7567	0.9848
T1	X1, Y1, Z1, X2, *…*, Z4	0.0912	1.8124	0.9917
T2	X1, Y1, Z1, X2, *…*, Z4	0.0736	1.4563	0.9946
T3	X1, Y1, Z1, X2, *…*, Z4	0.1516	3.3521	0.977
T4	X1, Y1, Z1, X2, *…*, Z4	0.1418	3.3136	0.9799
T1	T2, T3, T4	0.0798	1.4903	0.9936
T2	T1, T3, T4	0.1099	2.0871	0.9879
T3	T1, T2, T4	0.1073	2.0099	0.9885
T4	T1, T2, T3	0.077	1.558	0.9941

**Table 4 sensors-21-06173-t004:** Estimation of X and Y using different features.

	X			Y		
**Features**	**NRMSE**	**MAPE**	**R^2^**	**NRMSE**	**MAPE**	**R^2^**
A1, A2, A3, A4	0.1078	1.8595	0.9884	0.3335	6.7182	0.8888
X1, Y1, Z1, X2, Y2, *…*, Z4	0.0696	1.4419	0.9952	0.1369	2.6887	0.9813

## Data Availability

Not applicable.

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
