# Peer review of "An IoT-Based System for Monitoring the Health of Guyed Towers in Overhead Power Lines"

_sensors, 2021, doi:10.3390/s21186173_

Round 1
Reviewer 1 Report
The paper presents a complete framework for monitoring mechanical stresses and deformation of guyed towers remotely using several sensors and low-power, Wi-SUN-enabled nodes. It also presents an initial solution for using neural networks to estimate the tower structural parameters.
It is an interesting and complete solution that solves a clear problem and fits very well the areas of interest of the journal. Also, it has a very clear practical value even with prototypes.
However, I do have some concerns regarding the novelty of the proposed solution and some other suggestions for improving the paper.
Regarding the contributions of the paper, most of them are based on the use of the hardware, and the few novel ideas, such as the use of neural networks, are in a very initial state with few performance evaluation results. In this regard, I suggest the authors improve the introduction of the paper better highlighting the contributions of the proposed solution.
Also, the paper lacks a related work section. It seems that some related work was shown in the Introduction, but it only made it harder to read. I suggest rewriting part of the introduction and creating this related work section.
About the related work, the paper lacks a serious literature review of the area. For instance, the whole paper only cites 14 references. And the proposed solution is never compared to other solutions from the literature.
It would be good to also add some references to justify and contextualize the use of machine learn and neural networks. Also, make the input and output information of the neural network more clear (maybe a figure or diagram?)
The framework itself speaks a lot about the physical structure of the equipment and technologies that would be used and some things about the data flow and network topology, but it lacks some basis for the choice of the respective technologies.
It is not specified how the simulation data used to define the correlation was made using the model described. Is it stochastic? Was any simulator used? Is the data fully synthetic based on models?
In what follows, I'll show some other minor changes and suggestions to improve the quality of the paper in case it is accepted.
In the abstract, correct "we expected the risk of collapse of guyed cable towers reduces".
In line 103, Figure 6 is confused with Figure 4 in the sentence "Figure 6 illustrates a topology..." -> "Figure 4 illustrates a topology..."
Section titles should follow a pattern. In sections 2.1 and 2.2, the author alternates between upper and lower case letters for the initials of words. In other sections, all words start with an upper letter.
In section 2.2, line 122, the authors mention MCU for the first time without explaining what the acronym means.
I suggest a better explanation regarding the importance of each sensor used in the solution. The authors only mentioned the sensor's functions.
In line 195, the unit must have space after the value. "57,000Kgf"-> "57,000Kgf".
In line 282, the authors cite Figure 6 as the topology, but this figure is not about topology.
In line 328, the phrase "...computer interface 13 (a)..." should be written as "...computer interface depicted in Figure 13 (a)".
In section 3, Figure 11 should be cited as Figure 14. "Figure 11 illustrates simplified analytical" -> "Figure 14 illustrates simplified analytical". Again, in the same section, in line 344, the Figure 11 should be cited as Figure 14. "Notice, in Figure 11, that the..." -> "Notice, in Figure 14, that the...".
Equations 1 and 2 are not mentioned in the text.
In line 397, the phrase "...from the data set using (3)." should be written as "from the data set using Equation (3)."
The authors explain that the data were generated by simulation but do not explain the amount of data used.
The performance evaluation only takes into account the position, tension, and angle information, without mentioning the information from the other sensors.
Reviewer 2 Report
The paper is overall acceptable. Some minor changes should be made.
The abstract should be rewritten. Please, form the abstract in the following manner. First, describe the background of the research (1-2 sentences). Second, describe the goals of the research (1-2 sentences). Third, describe briefly (1-2 sentences) the methodology used. Fourth, describe the results and the conclusion of the research in 3-4 sentences.
The introduction is well written, other sections of the paper are described.
The results are well presented and interested for the readers.
Round 2
Reviewer 1 Report
This manuscript was improved if compared with the previous review round. In particular, all the comments of the reviewers were properly addressed in the current version.